# Reporting Quality of Randomized Controlled Trial Abstracts on Aesthetic Use of Botulinum Toxin: How Much Do Abstracts Actually Tell Us?

Ante Sucic [1], Ana Seselja Perisin [2,*], Tomislav Zuvela [2], Dario Leskur [2], Doris Rusic [2], Darko Modun [2] and Josipa Bukic [2]

1  Department of Otorhinolaryngology, Head and Neck Surgery, University Hospital of Split, 21000 Split, Croatia; asucic@kbsplit.hr
2  Department of Pharmacy, University of Split School of Medicine, 21000 Split, Croatia; tomislav.zuvela@mefst.hr (T.Z.); dario.leskur@mefst.hr (D.L.); josipa.bukic@mefst.hr (J.B.)
*  Correspondence: aperisin@mefst.hr; Tel.: +385-2155-7851

**Abstract:** Botulinum toxin use has become the most commonly performed aesthetic procedure among individuals of all age groups, encompassing both women and men. Randomized controlled trials (RCTs) provide the highest level of evidence and quality reporting of their abstracts plays a significant role for health professionals, influencing their decision-making in patient management. Therefore, our study aimed to assess the reporting quality of published RCT abstracts for botulinum toxin aesthetic use in the head area. The CONSORT-A checklist with 17 items was used to assess the quality of reporting. All available RCT abstracts (N = 191) found by searching the Medline database that were published up until June 2023 were included in this study. The average reporting rate was 52.9%. General items were inadequately reported (30.9%), with few abstracts describing the trial design. The methods section was better reported (62.0%), with interventions, objectives, and outcomes properly reported in over 97.5% abstracts. The results section (56.9%) demonstrated good reporting of randomized participant numbers but limited reporting of primary outcomes and harms. None of the abstracts reported funding sources or randomization information. To enhance the transparency and reliability of RCT results, abstracts should adhere more rigorously to the CONSORT-A guidelines. Improved reporting in abstracts can facilitate evidence-based decision-making in everyday practice of medical professionals in the field of aesthetic medicine.

**Keywords:** botulinum toxin; botox; cosmetic minimally invasive procedures; aesthetic procedures; randomized controlled trials; reporting quality; CONSORT

## 1. Introduction

Initially recognized as a toxin responsible for food poisoning, botulinum toxin has undergone a significant shift and now serves as an effective tool in aesthetic medicine, revolutionizing the way we approach facial rejuvenation. Since its discovery, botulinum toxin has been used in treating various disorders. The toxin's ability to block acetylcholine release at the neuromuscular junction offers relief to patients suffering from neuromuscular disorders, leading to the approval of Onabotulinumtoxin A for therapeutic use [1–4]. Numerous additional FDA approvals have been granted for the utilization of Onabotulinumtoxin A in various conditions, as well as for other products of botulinum toxin type A and a product of botulinum toxin type B [1]. In addition to Onabotulinumtoxin A (sold under the brand name of Botox), some of the commercially available botulinum toxin products used for medical and cosmetic purposes include Abobotulinumtoxin A (Dysport), Incobotulinumtoxin A (Xeomin), Prabotulinumtoxin A (Jeuveau), and Rimabotulinumtoxin B (Myobloc) [4].

Initially used for the temporary improvement of moderate-to-severe glabellar lines, botulinum toxin is now used in various aesthetic procedures. Some of the most common cosmetic procedures for which botulinum toxin is used include facial wrinkles' reduction, facelift, brow lift, neck bands, gummy smile, and masseter reduction [5–11].

The rising public awareness and acceptance of facial aesthetic enhancement have led to a growing demand for botulinum toxin treatments among individuals of all age groups, encompassing both women and men [12,13]. Over the years, botulinum toxin has become the most commonly performed aesthetic procedure, even in the group of patients aged 18 years or younger, overtaking rhinoplasty in previous years [14]. According to the American Society of Plastic Surgeons' statistical report for 2020, botulinum toxin type A was administered 4.4 million times for cosmetic minimally invasive procedures. In 2021, the number of botulinum toxin treatments was observed to have a significant increase of 17.7% compared to the previous year and an astounding 459% rise compared to the data from 2000 [14,15].

Several factors contribute to the growing popularity of cosmetic procedures. Patients seek not only to enhance their physical appearance but also to improve their mental and emotional well-being and overall social satisfaction. The desire for increased confidence, alleviating feelings of depression and anxiety related to their current appearance, and reducing the constant effort to conceal undesirable physical features are common motivations driving the pursuit of cosmetic treatments [16–20].

Given the ever-increasing popularity of botulinum toxin treatments in recent years, the role of randomized controlled trials (RCTs), as the highest level of evidence, becomes crucial in assessing the efficacy and safety of botulinum toxin injections. Abstracts play a significant role in the reporting of RCTs. They serve as a concise summary, containing essential study information that enables readers to quickly evaluate the study's originality, methodology, and results [21]. As the first section often read in a publication, the abstract significantly influences the readers' decision to proceed with reading further [21,22]. For health professionals, abstracts serve as a primary source of information about a trial, influencing their decision-making in patient management. In many cases, due to time constraints or limited access to full texts, practitioners rely solely on abstracts to make treatment decisions, making the quality of abstract reporting paramount [23,24]. Furthermore, researchers conducting systematic reviews and meta-analyses rely on abstracts to identify eligible studies, and poor reporting can lead to the exclusion of relevant research, potentially distorting evidence synthesis. Similarly, abstracts play a pivotal role in the literature databases, as indexers depend on adequate reporting to determine search terms and facilitate access to relevant articles [24].

The CONSORT (Consolidated Standards of Reporting Trials) addresses issues arising from inadequate data reporting in RCTs. Its central component, the CONSORT statement, provides evidence-based recommendations for transparent reporting of trial data. Introduced in 2008, CONSORT-A offers authors a comprehensive checklist, comprising 17 essential elements to be taken into account when presenting the RCT data in journal and conference abstracts [25,26].

These elements encompass identification of the study as an RCT in the article's title and description of the trial design to ensure proper database indexing, as well as inclusion of contact information for the corresponding author in case further clarification is needed. In the methods section, eligibility criteria and the setting wherein the data were collected, the interventions intended for each group, objectives and outcomes, participant allocation and randomization, and the use of blinding should all be clearly stated. These details aid readers in assessing the trial's validity and applicability of results. Sample size, number of participants analyzed in each group, trial status, outcomes, and adverse effects should be described in the results section, followed by a conclusion of the trial. The final two items in the CONSORT-A pertain to trial registration, serving to address selective reporting concerns, and the disclosure of funding sources, enabling the assessment of potential bias toward sponsors [23,27].

By adhering to these guidelines, authors can optimize the reporting of RCT abstracts, thereby facilitating better comprehension, evaluation, and application of trial results in medical practice. Studies conducted after the publication of the CONSORT guidelines for abstracts have examined the quality of published abstracts in various branches of medicine, including dermatology and plastic surgery, and mainly reported insufficient reporting quality [21–25,28–30]. Given the increasing demand for botulinum toxin treatments, the quality of published RCTs becomes essential. Therefore, the objective of this study was to evaluate the adherence of abstracts to the CONSORT-A statement on the aesthetic use of botulinum toxin in the head area and to investigate factors related to reporting quality.

## 2. Materials and Methods

The authors of this study conducted an observational study where they reviewed all abstracts from RCTs available in MEDLINE/PubMed about the topic of botulinum toxin use in aesthetic face treatments. In this study, RCTs were deemed eligible for inclusion if they featured a control group with participants allocated randomly. All types of RCT designs were eligible for inclusion. The included studies compared different treatments, including placebo, active treatment, or no treatment. The choice of outcome measures did not lead to the exclusion of any studies. Furthermore, RCTs with participants having comorbid diagnoses were also included in the analysis. In this study, certain types of abstracts were excluded from consideration. These omitted categories include non-clinical trials, observational studies without interventions, follow-up studies of previously published trials, reviews, protocols, as well as letters to editors and comments. This comprehensive approach ensured that the selected RCTs provided relevant and valuable data for the research objectives while maintaining the focus on clinical trials with interventions and control groups to yield reliable and meaningful findings.

In order to conduct the search on MEDLINE/PubMed, the following strategy was used: ("Botulinum Toxins, Type A [Mesh] AND (face OR head) [All Fields]") along with the filter for RCTs. The search was additionally limited to studies published in English and to human studies only. Since botulinum toxin is used for other indications in the head area besides aesthetic ones, the mentioned studies were excluded from further review and analysis (e.g., migraine or chronic headache, spasmodic dysphonia, pain reduction, blepharospasm, sialorrhea, allergic rhinitis, etc.). The search was carried out on 2 June 2023, and the complete list of extracted abstracts is available upon request to the authors.

All available RCT studies were included regardless of the year of publication. However, after extraction, the abstracts were divided into 2 groups according to the year of publication. Group 1 contained abstracts of RCT studies published before 2010, while Group 2 included studies published from 2010 to the date of the search. The year 2010 was selected as the dividing point for our search because the CONSORT for abstracts guidelines were published in 2008, allowing researchers conducting RCTs two years to incorporate these guidelines into their studies [23].

The determination of the quality of reporting in the included abstracts was achieved by assessing the authors' adherence to the guidelines listed in the CONSORT-A checklist. The CONSORT-A guidelines checklist provides 17 items that authors should follow when writing abstracts for their RCT study. Before reviewing the abstracts, the authors decided to assign a binary grade (0 or 1) to each of the mentioned items—if the reviewed abstract met the criteria for an adequately reported item, it was assigned the value 1, and if it did not, it was assigned a zero. The software program Microsoft Excel 2019 was used to collect the data, and the data mask was formed on the base of the CONSORT statement for abstracts [27]. The reporting quality of each abstract was determined by calculating a total score, a method that was used in previously published studies [23,30–32]. The reporting quality score was defined as the number of achieved items presented in each abstract, on a scale ranging from 0 to 17. The total score was presented as a whole number but also as a percentage of the number of items presented in each abstract in relation to the total number of items.

Additional factors were also included as potential predictors of the abstracts' reporting quality. The variables included in the analysis were as follows: study sample size (number of included participants <100 or ≥100), journals' impact factor and quartile, number of authors, type of abstract (structured or non-structured), presence of a recommendation for the use of CONSORT statement in the journal's instructions for authors, number of clinical centers (single or multi-centric study), statistical significance of the results (favoring experimental or control treatment), hospital setting, and funding by industry [23–25,33]. To determine the journals' impact factor and quartile, data from the Thomson Reuters Journal Citation Report for 2022 were utilized and included as variables in the analysis. For the primary outcome measure, a result was considered significant if the *p*-value was less than 0.05. If the primary outcome was favored in the treated group, the result was considered significant. As non-inferiority trials are intended to confirm that the effect of a new treatment is not worse than the effect of an active control, no statistical difference in comparison to the control group was considered as a significant result in the case of a non-inferiority trial design.

Two authors independently screened and evaluated the extracted abstracts to ensure unbiased evaluation. One of the authors was an experienced otorhinolaryngologist, a subspecialist in head and neck plastic and reconstructive surgery with knowledge in conducting RCTs, and the other was a qualified healthcare professional with a background in the field of biomedicine. Before initiating the PubMed database search and the subsequent review and evaluation of abstracts, the authors assembled to precisely review the guidelines outlined by the CONSORT statement for abstract evaluation [27]. In cases of disagreement between the two aforementioned authors, resolution was achieved through discussion with the third author, an experienced healthcare professional well versed in pharmacological sciences and possessing broad experience in conducting RCTs. This approach ensured the final selection of abstracts was accurate and unbiased.

The authors assessed interobserver agreement to determine how consistent their responses were when reviewing and rating the quality of the abstracts. Therefore, the Cohen κ coefficient (the kappa statistic), as the most commonly used statistic for this purpose, was determined. It was considered sufficient if the kappa point was higher than 0.6 [34]. The data were presented using various formats, including overall numbers and proportions (%), means and standard deviations (SDs), means and 95% confidence intervals (CIs), or medians and interquartile ranges (IQRs), as appropriate. To identify factors associated with higher reporting quality, linear regression analysis was performed [23,30,35]. Univariate analysis was performed for each variable, with the total quality score serving as a dependent variable. Subsequently, a multivariate regression analysis was conducted, incorporating factors significantly associated with a higher quality score in the univariate analysis ($p < 0.05$). Statistical analysis was performed using SigmaPlot (version 12.3 for Windows, Systat Software Inc., Chicago, IL, USA) and SPSS (version 16.0, IBM Corporation, Chicago, IL, USA).

## 3. Results

### 3.1. Characteristics of Included Abstracts

In this study, an outline of the search strategy and eligibility testing is presented in Figure 1. Previously described PubMed search results yielded 1712 studies, but after including additional filters to select English language papers and restricting the selection to only human studies, 1674 studies remained. Of the reported studies, only 259 were RCTs, which underwent further screening to ensure compliance with the inclusion criteria. A total of sixty-eight studies were excluded, with the primary reason being the application of botulinum toxin for non-aesthetic indications. These indications included migraine or chronic headache (*n* = 10), spasmodic dysphonia (*n* = 7), pain reduction (*n* = 6), blepharospasm (*n* = 6), sialorrhea (*n* = 4), and allergic rhinitis (*n* = 3). Additionally, 23 studies were excluded because they were not relevant to this topic and the application of botulinum toxin in those studies was not in the head area. Two studies were not RCTs but observational studies

without intervention, which is why they were excluded from further review. Furthermore, seven studies were excluded as they were letters to the editor or comments and therefore lacked available abstracts.

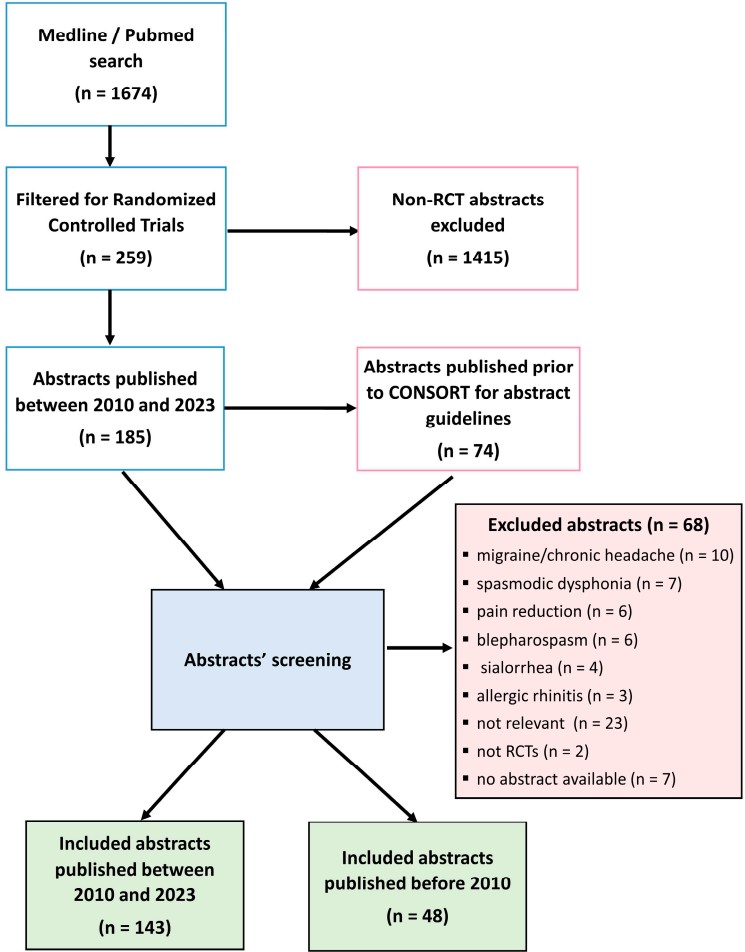

**Figure 1.** Search strategy and study selection.

Table 1 provides a detailed description of the study characteristics. Out of the 191 included abstracts, 122 (63.9%) were published in journals that endorsed the CONSORT guidelines for abstracts. Most of the abstracts were conducted in a single-center setting (133/191, 69.6%), and the majority of them included less than 100 participants (127/191, 66.5%). Notably, most of the reported results were statistically significant (164/191, 85.9%). Structured abstracts were predominant, accounting for 92.1% (176) of the included abstracts. The studies were published in journals with an average impact factor of 3.11 (SD = 2.49), and the median number of authors was 6 (IQR 3–8).

**Table 1.** Characteristics of included abstracts.

| Characteristics | N | % |
|---|---|---|
| **CONSORT endorsement** | | |
| No | 69 | 36.1 |
| Yes | 122 | 63.9 |
| **Published after 2009** | | |
| No | 48 | 25.1 |
| Yes | 143 | 74.9 |

**Table 1.** *Cont.*

| Characteristics | N | % |
|---|---|---|
| **Study centers** | | |
| Single center | 133 | 69.6 |
| Multicenter | 58 | 30.4 |
| **Significance of results** | | |
| Non-significant | 27 | 14.1 |
| Significant | 164 | 85.9 |
| **Number of participants** | | |
| <100 | 127 | 66.5 |
| ≥100 | 64 | 33.5 |
| **Funding** | | |
| Non-industry | 88 | 46.1 |
| Industry | 103 | 53.9 |
| **Setting** | | |
| Non-hospital | 149 | 78.0 |
| Hospital | 42 | 22.0 |
| **Abstract structure** | | |
| Unstructured abstract | 15 | 7.9 |
| Structured abstract | 176 | 92.1 |
| **Quartiles** | | |
| Non-ranked | 0 | 0.0 |
| 1st | 37 | 19.4 |
| 2nd | 40 | 20.9 |
| 3rd | 91 | 47.6 |
| 4th | 23 | 12.0 |
| | **Mean (SD)** | **Median (IQR)** |
| Number of authors | 6.93 (7.40) | 6.00 (3.00–8.00) |
| Impact factor | 3.11 (2.49) | 2.40 (2.30–2.90) |

*3.2. Quality Assessment of Individual Consort Items for Abstracts*

A comparison of the reporting results between the two authors showed that the lowest Cohen κ value for an individual item was 0.918 (minimum required 0.6). This confirmed the substantial interobserver agreement and fulfilled the main condition for further data analysis.

### 3.2.1. General Criteria Consort-A Items

The adherence of each item to the CONSORT for abstracts guideline is presented in Table 2. Among the reviewed abstracts, only 46.6% of them mentioned the term "randomized controlled" as the study design in the title (89/191). In 31.4% of the abstracts (60/191), contact details for the corresponding author were provided. However, only 14.7% of the abstracts (28/191) contained adequate descriptions of the trial design.

### 3.2.2. Methods Section Consort-A Items

The study methodology including interventions, objectives, and outcomes were well-reported in the majority of the abstracts, with 97.9% (187/191) adequately covering interventions, 99.5% (190/191) covering objectives, and 98.4% (188/191) covering outcomes. However, none of the reviewed abstracts provided a description of the randomization process. In 70.2% of the abstracts (134/191), blinding was mentioned, whereas only 5.8% of the studies (11/191) described the participants' inclusion criteria.

**Table 2.** Quality of individual CONSORT for abstract items.

| Items | N | % |
|---|---|---|
| Title | 89 | 46.6 |
| Authors | 60 | 31.4 |
| Trial design | 28 | 14.7 |
| **Methods** | | |
| Participants | 11 | 5.8 |
| Interventions | 187 | 97.9 |
| Objective | 190 | 99.5 |
| Outcome | 188 | 98.4 |
| Randomization | 0 | 0.0 |
| Blinding | 134 | 70.2 |
| **Results** | | |
| Numbers randomized | 173 | 90.6 |
| Recruitment | 190 | 99.5 |
| Numbers analyzed | 105 | 55.0 |
| Outcome | 19 | 9.9 |
| Harms | 56 | 29.3 |
| Conclusions | 190 | 99.5 |
| Trial registration | 15 | 7.9 |
| Funding | 0 | 0.0 |

### 3.2.3. Results Section Consort-A Items

Out of the 191 abstracts, 90.6% (173) included the number of participants randomized to each group in the study. However, only 55.0% (105/191) reported the number of participants included in the final analysis. In 74.9% of the abstracts (338/451), the primary outcomes were adequately reported, including both, effect sizes and its precision. Harms in RCTs, characterized by adverse events and side effects, were described in 56/191 (29.3%) abstracts.

### 3.2.4. Conclusion Section Consort-A Items

The meaningful conclusions were provided in nearly all abstracts (190/191, 99.5%). Among the 191 included abstracts, 15 (7.9%) included trial registry information, and none of the reviewed abstracts mentioned the sources of funding.

### 3.3. General Reporting Quality

The abstracts demonstrated a median of 9 (IQR 7–9) out of 17 (52.9%) properly reported items. However, none of the included abstracts reported all 17 items. The highest number of reported items was 12 out of 17 (70.6%), which was found in six abstracts (6/191, 3.1%). Contrariwise, the lowest number of reported items was 4 out of 17 (23.5%), achieved by only one abstract (1/191, 0.5%). The scores representing the general quality of reporting are presented in Table 3. A statistically significant difference was observed when comparing the total reporting quality score of abstracts published before 2010 and those published after. The earlier published abstracts scored higher (9.00) compared to the later ones (8.00). Additionally, the overall reporting quality score for each study characteristic is presented in Table 4.

**Table 3.** Overall reporting quality score.

| | Overall Score | Score (%) | Score before 2010 | Score (%) | Score after 2010 | Score (%) |
|---|---|---|---|---|---|---|
| **Mean** | 8.56 | 50.35 | 8.88 | 52.20 | 8.46 | 49.76 |
| **SD** | 1.52 | 8.98 | 1.38 | 8.11 | 1.56 | 9.18 |
| **95% CI** | 8.34–8.78 | 49.08–51.63 | 8.48–9.28 | 49.88–54.59 | 8.20–8.72 | 48.26–51.29 |
| **Median** | 9.00 | 52.94 | 9.00 | 52.94 | 8.00 | 47.06 |
| **IQR** | 7.00–9.00 | 41.18–52.94 | 8.00–10.00 | 47.06–58.82 | 7.00–9.00 | 41.18–52.94 |
| **Min** | 4.00 | 23.53 | 5.00 | 29.41 | 4.00 | 23.53 |
| **Max** | 12.00 | 70.59 | 11.00 | 64.71 | 12.00 | 70.59 |

**Table 4.** Overall reporting quality score for each study characteristic.

| Characteristics | Mean Score (%) | 95% CI |
|---|---|---|
| **CONSORT endorsement** | | |
| No | 50.47 | 48.32–52.62 |
| Yes | 50.29 | 48.67–51.91 |
| **Type of intervention** | | |
| Non-pharmacological | 41.18 | n/a |
| Pharmacological | 50.42 | 49.12–51.69 |
| **Study centers** | | |
| Single center | 49.67 | 48.22–51.12 |
| Multicenter | 51.93 | 49.32–54.54 |
| **Significance of results** | | |
| Non-significant | 49.89 | 46.51–53.27 |
| Significant | 50.43 | 49.03–51.83 |
| **Number of participants** | | |
| <100 | 49.65 | 48.09–51.22 |
| ≥100 | 51.75 | 49.51–53.99 |
| **Funding** | | |
| Non-industry | 49.87 | 48.05–51.68 |
| Industry | 50.77 | 48.95–52.59 |
| **Number of authors** | | |
| <4 | 50.59 | 48.20–52.98 |
| 4–7 | 49.13 | 47.54–50.72 |
| >7 | 52.16 | 49.06–55.26 |
| **Setting** | | |
| Non-hospital | 50.22 | 48.72–51.72 |
| Hospital | 50.84 | 48.38–53.30 |
| **Abstract structure** | | |
| Unstructured abstract | 49.41 | 45.76–53.06 |
| Structured abstract | 50.43 | 49.07–51.80 |
| **Impact factor** | | |
| <2.4 | 49.83 | 47.36–52.31 |
| 2.4–2.8 | 49.05 | 47.11–50.98 |
| >2.8 | 52.46 | 50.09–54.83 |
| **Word count** | | |
| <200 | 47.30 | 45.20–49.40 |
| 201–250 | 50.91 | 49.03–52.78 |
| >250 | 52.07 | 49.45–54.70 |
| **Quartiles** | | |
| Non-ranked | n/a | n/a |
| 1st | 53.74 | 50.88–56.59 |
| 2nd | 50.29 | 46.97–53.62 |
| 3rd | 48.93 | 47.19–50.67 |
| 4th | 50.64 | 47.06–54.22 |

*3.4. Reporting Quality Predictors*

Table 5 displays the results of the linear regression analysis. In the univariate model analysis, the number of words in the abstract were between 201–250 ($p < 0.05$) and more than 250 ($p < 0.001$). Additionally, the higher quartile of the journal was positively affected by the improved reporting quality of the RCT abstract. However, only a statistically significant association was found between publication at the first and third quartile of the journal ($p < 0.001$). The multicenter setting, significance of results, higher number of participants, funding by industry, hospital setting, a higher impact factor of the journal, and structured abstract positively correlated with a higher quality reporting score. However, a statistically

significant correlation was not observed for those predictors, and as a result, they were excluded from the multivariate analysis. In the multivariable model, a higher total quality score continued to be related to the number of words in the abstract between 201–250 ($p < 0.05$) and more than 250 ($p < 0.05$) and with publishing in a journal in the first quartile in comparison to the journal in the third quartile ($p < 0.05$).

**Table 5.** Linear regression derived estimates and 95% CI with a dependent variable defined as mean overall quality score shown as a percentage.

| Characteristics | Univariate Analysis, Estimate 95% CI | Multivariate Analysis, Estimate 95% CI |
|---|---|---|
| **CONSORT endorsement** | | |
| No | Reference | |
| Yes | −0.180 (−2.850 to 2.490) | |
| **Study centers** | | |
| Single center | Reference | |
| Multicenter | 2.259 (−0.512 to 5.029) | |
| **Significance of results** | | |
| Non-significant | Reference | |
| Significant | 0.539 (−3.141 to 4.220) | |
| **Number of participants** | | |
| <100 | Reference | |
| ≥100 | 2.094 (−0.607 to 4.794) | |
| **Funding** | | |
| Non-industry | Reference | |
| Industry | 0.905 (−1.665 to 3.475) | |
| **Number of authors** | | |
| <4 | Reference | |
| 4–7 | −1.457 (−4.573 to 1.658) | |
| >7 | 1.576 (−1.893 to 5.045) | |
| **Setting** | | |
| Non-hospital | Reference | |
| Hospital | 0.623 (−2.472 to 3.719) | |
| **Abstract structure** | | |
| Unstructured abstract | Reference | |
| Structured abstract | 1.023 (−3.743 to 5.788) | |
| **Impact factor** | | |
| <2.4 | Reference | |
| 2.4–2.8 | 0.788 (−3.917 to 2.340) | |
| >2.8 | 2.625 (−0.666 to 5.917) | |
| **Word count** | | |
| <201 | Reference | Reference |
| 201–250 | 3.609 (0.463 to 6.755) * | 3.605 (0.486 to 6.725) * |
| <250 | 4.774 (1.440 to 8.109) ** | 4.224 (0.740 to 7.708) * |
| **Quartiles** | | |
| 1st | Reference | Reference |
| 2nd | −3.442 (−7.425 to 0.541) | −3.458 (−7.488 to 0.392) |
| 3rd | −4.803 (−8.208 to −1.398) ** | −4.236 (−7.699 to −0.744) * |
| 4th | −3.097 (−7.723 to 1.540) | −2.278 (−6.998 to 2.443) |

* $p < 0.05$, ** $p < 0.01$.

## 4. Discussion

As the popularity of botulinum toxin treatments continues to grow, the significance of RCTs becomes crucial in assessing the efficacy and safety of botulinum toxin injections as the highest level of evidence. Within this context, abstracts play a vital role in reporting the findings of RCTs, and therefore, it is crucial that they follow the CONSORT-A guidelines.

To the best of our knowledge, this study represents the first assessment of RCT abstract reporting quality using the CONSORT-A checklist for the botulinum toxin aesthetic use in the head area. After reviewing 191 abstracts of botulinum toxin RCTs, we found that their general reporting quality could be considered as unsatisfactory. The median value of reported items was 9 (IQR 7–9), accounting for 52.9% of the checklist, which indicates that half of the reviewed RCT abstracts reported just a little more than half of the recommended items in the CONSORT checklist. These results correspond with previously published studies where general reporting quality varied from 40% to 54.3% [11,22–24,26,34].

General items were the worst reported group of the CONSORT-A items, scoring an average of only 30.9% of the total possible score within their group. A very low number of abstracts that described the trial design significantly influenced the result (14.7%). Methods items were the most adequately reported group of the CONSORT-A items, scoring 62.0% of the total possible score within their group. The most significant contributing items to this score were properly reported interventions, objectives, and outcomes, all of which had an average score of over 97.5%. However, principal methodological characteristics, such as eligibility criteria for participants, data collection settings, and randomization, were found to be poorly reported or completely missing, as only 5.8% of the abstracts included participant details, and none of the 191 abstracts described the randomization process. The results section items (score 56.9%) were less adequately reported but still comparable to the results observed in the methods section because more than 90% of the reviewed abstracts properly reported the number of randomized participants and recruitment. However, more than 90% of the reviewed abstracts clearly missed reporting the defined primary outcome, and harms were not reported in more than 70% of the reviewed abstracts. The lack of declaration of the source of funding in all of the reviewed abstracts and trial registration information observed in less than 8% were quite concerning information, particularly considering all abstracts were from RCT studies.

Our findings indicate that the introduction of the CONSORT-A guidelines did not lead to an improvement in the reporting quality of RCT abstracts on botulinum toxin aesthetic use. Moreover, a statistically significant higher overall quality score was observed in studies published in the pre-CONSORT period. Our findings align with prior studies indicating suboptimal adherence to the CONSORT guidelines for abstracts across various journals and medical disciplines, including dermatology and plastic surgery [21–25,28,29,34,36,37]. The purpose of adopting the CONSORT guidelines was to improve the quality of reporting, as it is necessary for the detailed translation of study results to clinicians' daily work. Therefore, journals that encourage the use of the CONSORT instructions were expected to have higher quality reporting in abstracts. However, this positive correlation was not present in our study. From all the mentioned sources, it appears that the adoption of the CONSORT-A guidelines has led to very little improvement in the quality of reporting in the abstracts or it has even been absent altogether, which is in line with the results of our research. The reason for such a result may be insufficient encouragement of authors by the journal to strictly recommend CONSORT-A. It was noted that most journals that recommend CONSORT do not provide additional instructions for the use of the CONSORT-A guidelines [37].

Funding source and randomization were found to be the lowest rated items according to the CONSORT guidelines, as none of the abstracts reviewed contained this information. The source of funding proved to be an inadequately reported item in RCT abstracts among published papers, as its value varied from 0 to 4 in several studies [21–23,38]. Ensuring transparency of funding sources is crucial, especially because industry funding is often associated with favorable outcomes, which could lead to readers' skepticism of research findings. The study by Lopez et al. showed that there is a positive correlation between industry funding and positive research outcomes in plastic surgery, but also in other branches of medicine where industry funding has often been biased [39–41]. Although the funding was not reported in any of the abstracts reviewed, the source of funding was extracted from the full paper. A minor difference of +0.9% in favor of higher quality

reporting in the abstracts of non-industry funded RCTs was observed. However, the difference was too small to be statistically significant.

According to the results of our study, a statistically significant correlation between a higher word count in the abstract and better quality of reporting was found. Hopewell et al., in presenting the CONSORT-A guidelines, specified that the abstract should be 250 to 300 words long to ensure that all items could be adequately reported [27]. Most studies have shown that the quality of reporting was better in abstracts with more than 250 words, because it is difficult to write all 17 required items in fewer words. However, most journals have their own rules and regulations, so the length and structure of the abstract varies from journal to journal. Therefore, it would be recommended that journals publishing RCT studies and recommending CONSORT guidelines not have a word count limit for abstracts of 200.

Impact factor and journal quartile showed a positive correlation with the quality of abstracts in many published RCT studies covering different areas of medicine [23,26,37,38]. The higher impact factor and the quartile of the journal indicate a higher quality of the journal and probably a stricter control of the reporting quality. However, in our study, higher abstract quality was positively correlated only with the first quartile of the journal, while the correlation with the impact factor was absent. However, the impact factors in different research fields are quite different and the studies on the aesthetic use of botulinum toxin are published in journals from dermatology, plastic surgery, otorhinolaryngology area etc. The range of impact factors is quite different from category to category, therefore, this correlation might be absent.

This study has some limitations. First, we limited our search to the PubMed search engine because it is the most accessible and widely used search engine among specialists in the field of biomedicine and healthcare. The MEDLINE database covers most of the biomedical literature, lifestyle journals, and online books and is the only one freely available to everyone, regardless of whether they are members of the academic community, biomedical specialists, or even patients themselves. The exclusion of Scopus as the broad database and Web of Science as the oldest database may have resulted in certain RCTs not being included in our analysis. Additionally, the impact factor of all journals in our study is stated for the year 2022, not for the year in which the paper was published. It is possible that the impact factor and journal quartile were different in the year the study was published, which could influence the results. However, our study possesses several strengths. First, we included all available summaries of RCTs, regardless of the year of publication, starting from the earliest available data on PubMed up to the date of the search. Additionally, as an indicator of the quality of the reviewing processes, the lowest Cohen's κ value for an individual item was 0.918, which represents a high degree of agreement among individual reviewers. Although some authors omitted the authors' contact as an item on the checklist, we decided to include it in the analysis. The reason for this is that although the authors always try to describe the research methodology and the results as good as possible, sometimes readers still have questions and doubts after reading. Therefore, we believe that having the option to contact the corresponding author is essential, as it allows for both addressing inquiries and gaining access to the complete data of the study.

## 5. Conclusions

Although CONSORT-A guidelines were published in 2008, there has been no improvement in the quality of reporting in RCT abstracts concerning the aesthetic use of botulinum toxin in the head area. While RCT studies represent the highest level of evidence, reporting in their abstracts has not been satisfactory. Inadequate reporting of the randomization process, selection of study participants, study design, reporting of adverse events, and clinical trial registration to a complete lack of reporting of the funding source, were most commonly observed in the reviewed abstracts. The poor quality of reporting in RCT abstracts and the specificity of missing data (e.g., funding information) increase mistrust in their credibility and reduce their potential usefulness for clinical decision-making. A

significantly higher quality of reporting was associated with a higher number of words in the abstract as well as a higher quartile of the journal in which the RCT study was published. This aligns with previously published studies, which all highlight higher word count as a key factor associated with better reporting quality of RCT abstracts. Following the results of this research, journals that publish the results of RCT studies should not strictly limit the number of words in abstracts to less than 250, as this could reduce the quality of their reporting and potentially negatively influence the applicability of the study results.

**Author Contributions:** Conceptualization, A.S., A.S.P. and D.M.; Methodology, D.R., J.B., D.L. and A.S.P.; Validation, D.L. and J.B.; Formal analysis, D.L., A.S.P. and A.S.; Investigation, A.S., A.S.P., T.Z. and D.M.; Resources, A.S., A.S.P. and T.Z.; Data curation, D.R. and J.B.; Writing—original draft preparation, A.S., A.S.P. and T.Z.; Writing—review and editing, D.L., J.B., D.R., A.S.P. and D.M.; Visualization, A.S. and A.S.P.; Supervision, D.M. and J.B.; Project administration, J.B. All authors have read and agreed to the published version of the manuscript.

**Funding:** This research received no external funding.

**Institutional Review Board Statement:** Not applicable.

**Informed Consent Statement:** Not applicable.

**Data Availability Statement:** Raw data is available from the corresponding author [A.S.P.] upon reasonable request.

**Conflicts of Interest:** The authors declare no conflict of interest.

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
