# Peer review of "Reporting Quality of Randomized Controlled Trial Abstracts on Aesthetic Use of Botulinum Toxin: How Much Do Abstracts Actually Tell Us?"

_cosmetics, doi:10.3390/cosmetics10050119_

Round 1

Reviewer 1 Report

This is interesting manuscript on quality of abstracts including Botox use for aesthetic purposes. I must add that I particularly enjoyed reading the introduction. The use of Botox in aesthetics has rise in the near past. Therefore, I believe this manuscript will be of interest to a broad audience, aesthetic practitioners, clinicians, patients and also scientists who will conduct future studies of Botox efficacy. 

I only have few suggestions for the authors:

1. Please add years of publications in the abstract 

2. Line 108 - you used tab in the introduction, do the same in other sections 

3. Line 169 - is a bolded?

4. Line 215 - additional space after 92.1

5. Line 285 - could you use not applicable instead of reference in table 5?

6. Line 292 - This abbreviation has already been introduced 

7. Line 296 - please use the introduced abbreviation

Author Response

Q1: This is interesting manuscript on quality of abstracts including Botox use for aesthetic purposes. I must add that I particularly enjoyed reading the introduction. The use of Botox in aesthetics has rise in the near past. Therefore, I believe this manuscript will be of interest to a broad audience, aesthetic practitioners, clinicians, patients and also scientists who will conduct future studies of Botox efficacy.

A: Dear reviewer, we are grateful for your comments on our manuscript.

Q2: Please add years of publications in the abstract.

A: Thank you for this important observation. As all available RCT studies were included regardless of the year of publication, we have added only the last year and month (time of search) in sentence. The sentence has been rewritten rewritten as follows: “All available RCTs abstracts (N=191) found by searching Medline database and published until June 2023 were included in this study.”

Q3: Line 108 - you used tab in the introduction, do the same in other sections

A: Thank you for this important observation. Suggestion fully accepted. Tabs were added to all sections in the manuscript.

Q4: Line 169 - is a bolded?

A: Thank you for your comment. This was a typo made during writing and we have unbold the letter “a”.

Q5: Line 215 - additional space after 92.1

A: Thank you for your comment. The additional space after 92.1 has been deleted.

Q6: Line 285 - could you use not applicable instead of reference in table 5?

A: Thank you for your comment. However, we cannot fully accept your suggestion. The reason is that “reference” in the table 5 represents the variable (value) that is the reference, i.e. serves as the value which is compared with all other variables in that data set. Therefore, “not applicable” cannot be an appropriate replacement word for an existing word “reference” in univariate model. In multivariate analyses, word “reference” was deleted from the rows in table 5 where analyses was not possible to perform, and these rows are marked with gray background, as this is the most usual way of presentation.

Q7: Line 292 - This abbreviation has already been introduced

A: Thank you for this important observation. It has been deleted and abbreviation was used instead of full word.

Q8: Line 296 - please use the introduced abbreviation.

A: Thank you for your comment. Abbreviation was used instead of full words in line 296 but also in the whole manuscript.

Reviewer 2 Report

The manuscript "Reporting quality of randomized controlled trials abstracts on 2 aesthetic use of botulinum toxin: How much do abstracts actually tell us?" aims at evaluating the quality of published randomized control trials on botulinum toxin. It is a theoretical and data processing study in the area of cosmetic product.

The manuscript provides a detailed statistical analysis of data and the analysis results seem to be convincing. The overall impression of the manuscript is that it can be considered for publication in the topical collection "Feature Papers in Cosmetics in 2023" after addressing the following comments:

1) In the abstract, it is recommended to briefly outline the main findings of the article after stating its objective.

2) The discussion seems to be quite long for a smooth readability, although it is structured well and also critical. The authors are recommended, if applicable, to add a short conclusion where they can summarize the Discussion section

3) Are there any competitive techniques of statistical analysis, which can give alternative results based on the initial data considered? In this respect, what are particular benefits of the method used in this paper?

4) Although the language quality is good enough to render the science clear, a certain stylistic editing of English can improve the manuscript.

A certain minor stylistic editing of English can improve the manuscript

Author Response

Reviewer 2

Q1: The manuscript "Reporting quality of randomized controlled trials abstracts on 2 aesthetic use of botulinum toxin: How much do abstracts actually tell us?" aims at evaluating the quality of published randomized control trials on botulinum toxin. It is a theoretical and data processing study in the area of cosmetic product.

The manuscript provides a detailed statistical analysis of data and the analysis results seem to be convincing.

A: Dear reviewer, we are grateful for your comments on our manuscript.

Q2: In the abstract, it is recommended to briefly outline the main findings of the article after stating its objective.

A: Dear reviewer, thank you for your recommendation. However, it can not be completely accepted as the abstract was written according to the instructions for authors in Cosmetics, which state that the abstract must consist of the background (where the aims/objectives have to be described), methods, and after that results and conclusion. If we add the main findings of the study just after its objectives, our abstract will not follow the suggested abstract structure as written in the instructions for authors.

Q3: The discussion seems to be quite long for a smooth readability, although it is structured well and also critical. The authors are recommended, if applicable, to add a short conclusion where they can summarize the Discussion section.

A: Thank you for your valuable suggestion. Suggestion fully accepted. The conclusion was added at the end of the manuscript to summarize the Discussion section as follows: “Although CONSORT-A guidelines were published in 2008, there has been no improvement in the quality of reporting in RCTs abstracts concerning the aesthetic use of botulinum toxin in the head area. While RCT studies represent the highest level of evidence, reporting in their abstracts has not been satisfactory. Inadequate reporting of the randomization process, selection of study participants, study design, reporting of adverse events, and clinical trial registration, to a complete lack of reporting of the funding source, were most commonly observed in reviewed abstracts. The poor quality of reporting in RCTs abstracts and the specificity of missing data (e.g., funding information), increase mistrust in their credibility and reduce their potential usefulness for clinical decision-making. A significantly higher quality of reporting was associated with a higher number of words in the abstract, as well as a higher quartile of the journal in which the RCT study was published. This aligns with previously published studies, which all highlight higher word count as a key factor associated with better reporting quality of RCT abstracts. Following the results of this research, journals that publish the results of RCT studies should not strictly limit the number of words in abstracts to less than 250, as this could reduce the quality of their reporting and potentially negatively influence the applicability of the study results.”

Q4: Are there any competitive techniques of statistical analysis, which can give alternative results based on the initial data considered? In this respect, what are particular benefits of the method used in this paper?

A: Thank you for your comment. We have read many similar publications and went through statistical analysis descriptions to notice all the statistical methods used in these studies. Some publications used simple t-test or ANOVA for statistical analysis instead of regression models. However, the flexibility of the regression models as these we used in our study, allows us to perform most analyses using a unified approach. Using linear regression instead of a t-test or ANOVA allows us to directly obtain estimates (differences between treatment groups) along with their confidence intervals instead of only P values. This could be considered as the most important benefit of the regression models we used in comparison with the t-test or ANOVA.

Q5: Although the language quality is good enough to render the science clear, a certain stylistic editing of English can improve the manuscript.

A: Dear reviewer, we are grateful for your comment. Suggestion fully accepted. The manuscript has now been carefully read and reviewed by a native English speaker before submission. We have improved our manuscript accordingly and the changes are made by using the “Track changes” function.
